optics/biomaterials

spider silk, reflectance, scattering, absorption, dragline

**Author for correspondence:**
Deb M. Kane
e-mail: deb.kane@mq.edu.au

†Present address: Department of Physics and Astronomy, Macquarie University, Sydney, NSW 2109, Australia.

# Photoreflectance/scattering measurements of spider silks informed by standard optics

Sean J. Blamires[1], Douglas J. Little[2], Thomas E. White[3] and Deb M. Kane[2,†]

[1]Evolution and Ecology Research Centre, School of Biology, Earth and Environmental Sciences, University of New South Wales, UNSW Sydney NSW 2052, Australia
[2]Macquarie University Photonics Research Centre and Department of Physics and Astronomy, Macquarie University, Sydney NSW 2109, Australia
[3]School of Life and Environmental Sciences, The University of Sydney, NSW 2006, Australia

SJB, 0000-0001-5953-3723; TEW, 0000-0002-3976-1734;
DMK, 0000-0003-4551-9896

The silks of certain orb weaving spiders are emerging as high-quality optical materials. This motivates study of the optical properties of such silk and particularly the comparative optical properties of the silks of different species. Any differences in optical properties may impart biological advantage for a spider species and make the silks interesting for biomimetic prospecting as optical materials. A prior study of the reflectance of spider silks from 18 species reported results for three species of modern orb weaving spiders (*Nephila clavipes*, *Argiope argentata* and *Micrathena Schreibersi*) as having reduced reflectance in the UV range. (Modern in the context used here means more recently derived.) The reduced UV reflectance was interpreted as an adaptive advantage in making the silks less visible to insects. Herein, a standard, experimental technique for measuring the reflectance spectrum of diffuse surfaces, using commercially available equipment, has been applied to samples of the silks of four modern species of orb weaving spiders: *Phonognatha graeffei*, *Eriophora transmarina*, *Nephila plumipes* and *Argiope keyserlingi*. This is a different technique than used in the previous study. Three of the four silks measured have a reduced signal in the UV. By taking the form of the silks as optical elements into account, it is shown that this is attributable to a combination of wavelength-dependent absorption and scattering by the silks rather than differences in reflectance for the different silks. *Phonognatha graeffei* dragline silk emerges as a very interesting spider silk with a flat 'reflectance'/scattering spectrum which may indicate it is a low UV absorbing dielectric micro-fibre. Overall the measurement emerges as having the potential to compare the large numbers of silks from different species to prospect for those which have desirable optical properties.

# 1. Introduction

The sense of sight is common across the animal kingdom and relies on detecting, imaging and processing light reflected and scattered from surrounding surfaces. Basic tasks such as foraging, seeking mates and avoiding predators are often mediated visually [1,2], and the utility of light as a mode of information exchange has driven the evolution of optical innovations for manipulating it. A diverse range of spiders has evolved the construction and the use of webs for prey capture in parallel with the evolution of silks and other elements of the webs [3–5]. The mechanical structure and properties of many of the silks are well characterized but the aerial nature of some of the orb webs has allowed spiders to exploit ecological niches in well-lit, free space. In this case, the optical behaviour of the web, in addition to its mechanical behaviour, becomes highly relevant. Despite this, the optical properties of such silks are comparatively unknown. The birefringence of spider silks has been a topic of study because it correlates with the mechanical properties of silk [6–8]. Furthermore, the potential for a spider silk to be used as a micro-optical fibre [9] including in sensing applications [10] have been reported.

For spiders that have evolved to occupy aerial locations, especially locations of high brightness during the day, it is underappreciated that the silks of ecribellate orb webs (i.e. sticky prey capturing wheel-shaped webs) mostly comprise transparent materials of high optical quality [11–17]. The species *Argiope keyserlingi* and *Plebs eburnus* have been studied and shown to have high optical quality dragline/radial (i.e. major ampullate) silks [11–17]. The orb webs of adult ecribellate spiders have radial and capture silks with widths from one to a few micrometres. Thus, small size is one mechanism by which the amount of light reflected, or more correctly scattered, is reduced. A small physical footprint is presented by the silk to the incident light. Also, by virtue of being largely transparent, most of the light incident on a silk is transmitted into the silk and then undergoes reflection at the silk–air interface before being transmitted and scattered out of the silk. A first model for such a spider silk, as an optical element, is a transparent cylinder. In fact, the silks are typically double cylinders lying side by side [16,18]. Thus, in measuring the optical function of spider silks, it is important to take their form into account. They are better modelled, and thought of, as scatterers of light rather than reflectors. Herein we critically appraise photoreflectance measurements of spider silks, using a standard technique, taking into account the actual form of silk as an optical component. Useful information about the relative optical properties of the silks of different species are learned by this measurement, but correct interpretation requires the underlying optics to be included.

A comprehensive study of the relative reflectance spectra of spider silks and capture threads from a large number of different species was previously done by Craig *et al*. [19]. This study identified significant contrasts in the reflectance spectra, measured for the wavelength range 350–700 nm, for species that occupy different ecological niches. Broadly, the results were characterized into three categories: (i) spectra with enhanced reflectance in the ultraviolet (UV), (ii) flat spectra, and (iii) spectra with reduced UV reflectance. The study examined the silks of some derived ecribellate species (*Nephila clavipes, A. argentata* and *Micrathena Schreibersi*) that had normalized reflectance in the ultraviolet that was lower than that in the visible [19]. These spiders place their aerial webs in open habitats, such as canopy gaps and corridors, where any extra reflectance might make the webs become conspicuous to predators and prey. Normalized reflectance values do not give insight into the relative reflectance signal strength from the silk of different species. It is this latter information that would indicate which silks reflect more of the light which falls on them.

In this study, we measured the photoreflectance/scattering of major ampullate (MA) silks from four species of spider: *N. plumipes, Eriophora transmarina, A. keyserlingi* and *Phonognatha graefei*. All these spider species build an aerial orb web. All are found locally around Sydney, Australia. We note that *N. plumipes* is a congener of *N. clavipes* and *A. keyserlingi* is a congener of *A. argentata*, when making comparisons to the results of Craig *et al*. [19]. *Nephila plumipes* (or *Trichonephila plumipes* according to the nomenclature of Kuntner *et al*. [20]) is of the family Nephiliidae. It is primarily diurnal; however, many *Nephila* spp. forage by both day and night [21], which is likely but unrecorded for *N. plumipes*. Craig *et al*. [19] reported that *Nephila* spp. MA silk has relatively low reflectance in the UV compared to other wavelength bands. The three remaining species are from family Araneidae. *Argiope keyserlingi* is a diurnal forager that has been mostly attributed as building a new web daily at sunrise and dismantling it at sunset. Our observations indicate that a single web may be deployed for several days before being replaced. Its dragline (MA) silk would be thought to be of relatively low reflectance within the UV wavelength band, presumably to reduce its visibility to insects, as concluded for the similar species studied in [19]. Nonetheless, it commonly decorates its web with a conspicuous cruciform-shaped mass of aciniform silk that is

highly reflective in the UV [22,23]. *Eriophora transmarina*, is a large bodied spider. It builds a very large web with a capture spiral that may exceed 1 m in width. It is primarily nocturnal, although it does occasionally retain its web during the day. Being phylogenetically related to *A. keyserlingi*, we might expect its MA silk to likewise have low reflectance values in the UV wavelength band compared to other wavelength bands. Nonetheless, if silk visibility by day is the selective driving silk reflectance, we might expect *E. transmarina* silk to be slightly more reflective in the UV than that of *A. keyserlingi*. *Phonognatha graefei*, or leaf curling spider, is reported as a nocturnally active web building spider of the family Araneidae (but formerly classed with the more ancient family Tetragnethidae). Our observations indicate that this spider is active by day in suburban Sydney. A defining characteristic of its web is the inclusion of a rolled leaf into which the spider retreats [24]. As a member of the Araneidae, we might expect the MA silk of *P. graefeii* to exhibit relatively low reflectance in the UV wavelength band [21]. Phylogenetic information for the spiders in the study have been updated quite recently [21,25,26].

In studies of the reflectance of spider silks completed to date, the optics of the silk has not been considered in detail. A study of webs under UV light [27] did note that the measurements in [19] may have been affected by the form of the silks in ways that had not been taken into account. The silk of *A. keyserlingi* has a measured refractive index of approximately 1.55 [14–16] and has low absorption in the visible [17]. Treating it as a homogeneous optical material, i.e. a dielectric, its reflection coefficients at the air–silk interface will be governed by the Fresnel equations [28]. There is no wavelength dependence for these equations, other than a weak dependence of refractive index on wavelength. The measured refractive index of the *A. keyserlingi* MA silks is larger at shorter wavelengths compared to longer ones [14,15], and therefore, any small effect of this would increase the reflectance for the silks at UV wavelengths compared to visible wavelengths. Thus, there will be no expectation of a reduced reflectance at shorter wavelengths based on reflectance at a single, flat interface for a dielectric with a refractive index of approximately 1.55. One way that such a reduction might result is if there was a differentiated layer of a lower refractive index at the outer surface of the silk. Such a layer could act as an antireflection (AR) coating with a centre wavelength determined by the thickness of the layer. The centre wavelength could be in the UV. The aqueous layer on a capture silk could potentially form such an AR coating, for example. But, treating the silk as a flat transparent reflector, there is no physical basis to explain a reduced reflectance in the UV as compared to that at longer wavelengths. Any observed reduction is not able to be explained as a reflectance effect.

## 1.1. Silk optic—shape effects

The silks are not flat, transparent reflectors and so the effect of the shape of the silk on the effective reflected (backscattered) light has to be considered. A single, transparent cylinder of refractive index approximately 1.536 has been described as a first-order model for ecribellate radial silk [13]. This is a standard, light scattering problem that has been solved [29,30]. By virtue of the small size of the silks, the light is scattered into a broad range of angles. A comparison of the backscattered light from a glass cylinder with a diameter of 125 µm, $n = 1.456$, wavelength 633 nm, which is a size similar to an optical fibre (with no jacket), as presented in [13] and contrasted with a model of a spider silk. A spider silk with a diameter to 2 µm, $n = 1.536$, scattering light of wavelength 633 nm had an angular spread of the central backscattered peak of approximately 18° [13]. When it is considered that only the light that falls within the small footprint of the silk is partially backscattered, this shows that the advantage of small size is twofold—small collection footprint and redirection of the backscattered light into a broad range of angles.

Using the standard solution for scattering by a single cylinder [29], the pattern of light scattered by the cylinder through the full angular range of 360° is shown in figure 1 for a cylinder of radius 2 µm for wavelengths 350 nm (left) and 633 nm (right). The simulation uses a plane wave incident upon a dielectric cylinder at normal incidence, i.e. a horizontal plane wave propagating from the top towards the cylinder. To get the full picture of the forward-directed light after the silk, the forward-scattered light shown in the figure (angles 90° to 270°) needs to be combined with the rest of the input field that has been propagated. In the case of the backscattered light, as it is propagating in the opposite direction to the input light, the scattered light intensity is a good visualization of the light that can potentially reach a detector in a photoreflectance measurement.

The primary approach taken in this work is to study what results are obtained using a simpler, but essentially analogous, experiment to that reported in [19], as supported by current, commercially available photoreflectance measurement equipment. Such equipment has been used to measure the reflectance spectra of butterfly wings [31,32] and spider silk from the species *P. clavis* [33]. The latter is a

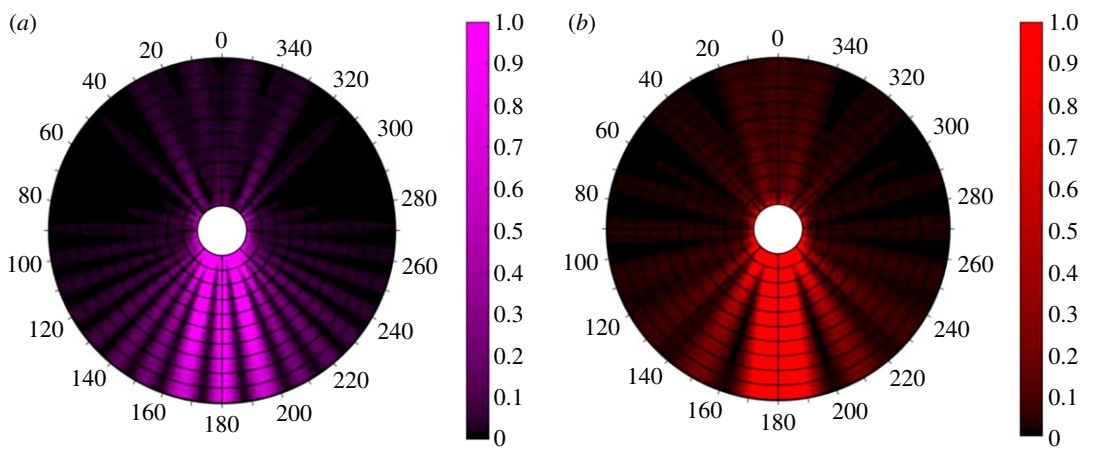

**Figure 1.** Modelled scattered light intensity for a 2 µm diameter single cylinder, $n = 1.536$, at wavelengths of 350 nm (*a*) and 633 nm (*b*). The diameter of the modelled region is 14 µm. The angle in degrees is annotated at the edge of the circular region. Zero degrees corresponds to centre of the backscattered light and 180° to the centre of the forward-scattered light. The computation for 2 µm was truncated at 30 cylindrical harmonic modes and that for 6 µm to 80 modes. The intensity map is an average of scattered light polarized parallel and perpendicular to the cylinder axis. The intensity scale is normalized to that of the incident wave and the colour scale allows lower level intensities values to be visualized.

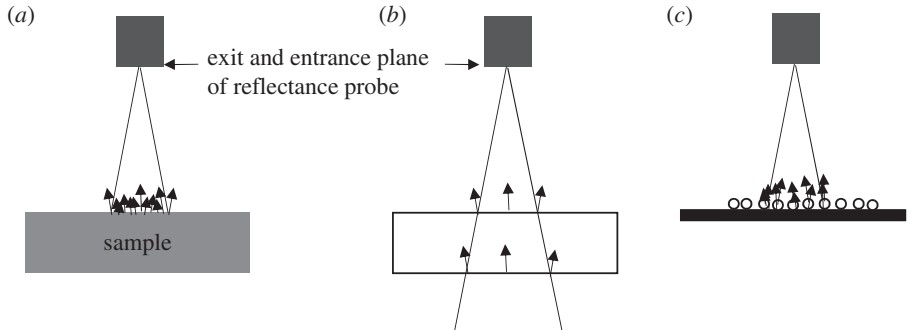

**Figure 2.** A schematic of the illumination of different sample types in reflectance spectroscopy. Only the end of the reflectance probe is shown. This probe is connected to the light source and the fibre-coupled spectrometer (not shown). The triangle with its apex at the exit/entrance plane of the probe indicates the cone of the angular spread of the light. (*a*) A solid colour sample reflecting light diffusively from the top surface only. (*b*) A polished glass sample which transmits most light but has specular reflection from the top and bottom surface (Fresnel reflection). (*c*) Transparent cylinders on an absorbing substrate which scatter, and transmit and scatter, the light.

cribellate silk that appears white when viewed by human vision or photographed. The latter study used a sample obtained by winding lots of layers from webs onto card. About 1 mm thickness of the silk was used [33]. Using a sample of the form used in [33] could potentially lead biologists to make measurements on a similar sample made by winding many layers of transparent, ecribellate silk. This study explains why such an approach is not appropriate. Before introducing the experiment and the results, the following information introduces the background optics relevant to informing the experimental design and the interpretation of the results.

## 1.2. Photoreflectance measurement—general background

Photoreflectance is a measurement that aims to give the spectrum of light that is reflected from a surface when it is illuminated with a standard white light source. It is most appropriately applied to a flat but diffusely reflecting single surface of solid colour. Any light that is not reflected from the surface is absorbed. Figure 2*a* shows a schematic of this situation. The light delivered to the sample from the optical fibre of the photoreflectance probe is diffusely reflected from the surface which is rough in an optical sense. The reflected light has a spectral content determined by the colour and properties of the surface, once corrections for the spectral content of the illuminating light are made. Some fraction of

the reflected light, the same fraction for all wavelengths, is collected within the numerical aperture of the collection core within the fibre optic of the photoreflectance probe (this collection is not illustrated). A spectrometer can then disperse this light onto a charge-coupled device (CCD) and the reflectance spectrum is captured electronically from the readout.

If the sample is partially transparent then light is scattered at different depths within the sample. If the sample is a plate of a high-quality glass as indicated schematically in figure 2b then a strong specular reflection occurs at the top surface and a second one at the bottom surface. The light that is not reflected is transmitted. For a glass window (refractive index $n = 1.55$) and for normally incident light, the transmitted light is about 91% while the reflected light is approximately 9% (taking both surfaces into account). If a range of angles of incidence are included, the transmitted light would drop to a lower value while the reflected light would increase, these two summing to 100% in the absence of any absorption. The Fresnel equations [28] allow these percentages to be calculated. The reflectance in this case is specular for a polished glass rather than diffuse as obtained from a roughened surface.

Figure 2c is a schematic of a cross-section of several transparent cylinders on a non-reflective (fully absorbing) surface. This is the case that is relevant to measuring the photoreflectance from many spider silks wrapped around an absorbing sample holder. In this case, light is scattered from the cylinders—both from the top surface and from the internal surface after transmission into the cylinder. Cylinders on the surface within the numerical aperture of the optical-fibre-delivered light are illuminated. The spatial variation of the scattered light for a narrow range of wavelengths will be similar to those shown in figure 1. Backscattered/reflected light that falls within the numerical angle of the collection optical fibre contributes to the light that is detected and analysed as the 'reflected light'. The strength of this signal can change with wavelength because of the geometric distribution of the scattered light changing. This effect can lead to a drop in 'reflectance' as a function of wavelength. In the study, here we are using a different technique to that used in [19] to repeat the measurement of the wavelength dependence of what can be interpreted as the signal an insect would detect as it flew towards an orb web. We show that two of the four species studied do have a reduced scattered light signal as a function of wavelength and that this should not be interpreted as a reduced reflectance. There is no physical basis for a reduction in reflection. With a micro-optic of the form of spider silk, it is not appropriate to refer to reflection as a valid physical quantity to be measured.

# 2. Material and methods

## 2.1. Spiders—collection

We collected six females (identified visually) of each of the four species of spider (N. plumipes, E. transmarina, A. keyserlingi and P. graeffei, totalling 24 spiders) from locations in suburban Sydney and taken to the laboratory at the University of New South Wales in eastern Sydney for silk collection. The six were a combination of non-gravid adult and sub-adult spiders.

## 2.2. Method of sample preparation

Aluminium strips approximately $80 \times 10 \times 1$ mm were cut. The edges were smoothed and the strips were cleaned and dried. A strip of black paper (approx. $60 \times 9$ mm) was glued to one side. This width was such that it remained clear of the edges of the aluminium strip and approximately 20 mm of aluminium remained exposed at the base of the strip to enable handling.

We collected silk from each of the 24 spiders onto the aluminium strips as follows. We first anaesthetized each of the spiders collected, using carbon dioxide, and placed them ventral side up on a foam platform and immobilized them with non-adhesive tape and pins. We then carefully collected an MA silk thread from the spinnerets and wound it once around an aluminium strip (see above), which was connected to an electronic spool rotating at 1 m per minute. The silk was initially wound around the strip at about 15 mm from the base of the black paper. We thenceforth slowly moved the spider's platform by hand so that silk reeled sequentially along the remaining length of the black paper. Since the platform was moved by hand the spacing between the individual rounds was not even.

We viewed the silking procedure under a dissecting microscope in an attempt to make sure that one thread of silk was consistently being collected. Once we had collected silk from all spiders onto the aluminium strips, the silk samples were individually placed into a sample tray with the aluminium

handle embedded within the holding cavity. They were subsequently transported within a sealed container to the Department of Physics at Macquarie University in northern Sydney.

## 2.3. Method for photoreflectance measurement

The photoreflectance measurements were carried out using a reflectance probe (Ocean Optics QR-400-7-UV-VIS) connected to a pulsed xenon light source (Ocean Optics PX-2). This fibre optic reflectance probe combines light delivery to the sample via six cores in the optical fibre arranged in a ring and light collection for spectral analysis via a central core in the optical fibre. All the optical fibre cores are 400 μm in diameter and have a numerical aperture of 0.22. This means the angular spread of the light as it leaves the optical fibre is about 25°. The spot size of the light source on the sample is determined by the distance between the optical fibre end facet and the sample. This is one means used to control the maximum intensity detected to ensure that the detector is not saturated. A larger distance increases the spot size and reduces the intensity on the sample. The acceptance angle for reflected light into the central fibre core is also 25°.

The collected reflected light is an optical fibre coupled to a spectrometer (Ocean Optics USB-4000, 200–1100 nm, resolution less than 2 nm) to generate the reflectance spectrum. The optical fibre and spectrometer combined detect light in the wavelength range 200–850 nm. The wavelength range of interest in this study is 300–700 nm. The software application Spectrasuite® was used to computer control the spectrometer and the collection of reflectance data. The reflectance values are corrected for variations in the intensity of the pulsed light source as a function of wavelength by the software. The light source was operated for a warm-up period before measurements began. Zero reflectance was set as the spectrum of the black paper and 100% reflection was referenced to the reflectance of a diffuse white reflector standard (Spectralon®, Labsphere). A check that the detector is not saturated by the light level from the white standard was made. Regular recalibration of the reference levels is good practice when making measurements. Any impact of drift in the reflectance spectra of the black and white reference needs to be noted and avoided. Care was also taken to ensure the black and white reference surfaces were at the same distance from the optical fibre end as the sample surface to be measured, when the reference spectra were captured. The sample was placed at a distance of approximately 9 mm from the end face of the optical fibre reflectance probe giving a spot size on the sample of about 4 mm. The end face of the probe was parallel to the sample plane (normal incidence). When first collecting spectra, an integration time of 10 ms, a boxcar of width 4 and the average of 50 scans were used for all four types of silk. This integration time is shorter than has been used in earlier studies [31,32]. It leads to the reflectance spectra being less smoothed. The variations in reflectance occurring for small changes in wavelength were due to fluctuations in the equipment rather than being real variations.

It was found that a total 'reflectance' signal level of the order of 4% led to a signal-to-noise level that was sufficient to give reproducible spectral results that could be attributed to the optical characteristics of the specific silk. This required a sufficient number of silks to be illuminated within the approximately 4 mm diameter illuminated spot on the sample to achieve this reflectance signal. The value of reflectance scaled as expected with the increasing number of silks illuminated, and reflectance values of up to 13% were obtained for *E. transmarina*. It is noted that it is the silks of this nocturnal species that have the largest photoreflectance signal per silk in this study. This is primarily because of their larger diameter. The normalized reflectance spectrum, normalized to the value of the reflectance at 500 nm, did not change qualitatively with increasing signal level. An improved signal to noise was achieved at higher signal level. A second set of samples needed to be prepared for *P. graeffei* with a larger number of threads per unit length than for the other three species to achieve the signal level required for meaningful measurement. The *P. graeffei* silks are thinner than the others. An integration time of 50 ms, a boxcar of width 5 and the average of five scans were used for the *P. graeffei* results that are reported.

The reflectance probe was held in a reflectance probe holder (RPH-1, Ocean Optics). This allows the probe to be held in a vertical position or at 45°. The measurements reported were made with the probe in the vertical position. Tests were carried out using *A. keyserlingi* and *N. plumipes* silk samples to ensure that the qualitative form of the reflectance spectra of the silks for light incident at 45° did not change. That this was observed is more evidence that the results are primarily due to the shape and high transparency of the silks. The silks are sufficiently far apart in the samples to avoid any effects of shadowing at higher angles of incidence. Consider tipping the sample shown in figure 2c to a larger angle. This does not change the backscattered light other than as due to a change in intensity associated with an oval rather than circular light spot on the sample.

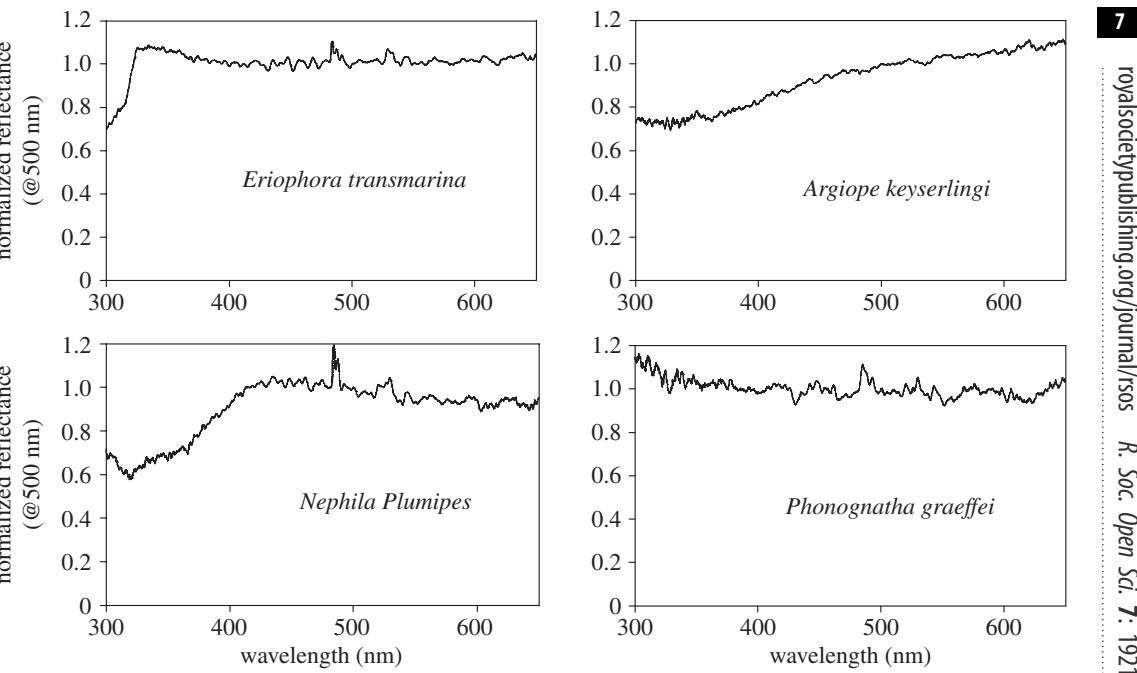

**Figure 3.** The normalized, averaged 'reflectance' spectra for the dragline silk from four species of orb web weaving spiders. The sharp peaks at approximately 485 nm and approximately 530 nm are artefacts. This is explained in the text.

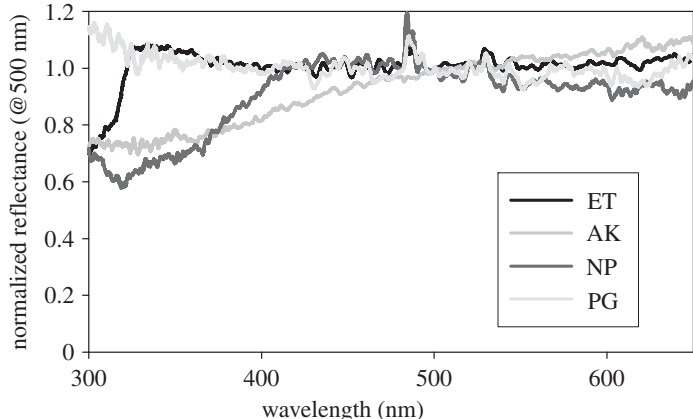

**Figure 4.** The results from figure 3 overlaid for direct comparison.

## 3. Results and discussion

The averaged normalized reflectance spectra for a single sample of silk for each of the four species are shown in figure 3. These spectra are overlaid in figure 4 for ease of relative comparison. The spectra are plotted for the wavelength range 300–650 nm. The raw data are provided in Excel spreadsheets as electronic supplementary material. Note that the individual reflectance spectra, and the average of several individual normalized reflectance spectra, all have essentially the same functional form. Averaging does not change the functional form. Given that the PX-2 pulsed xenon light source has an intensity at wavelengths of approximately 300 nm and approximately 700 nm that is only about 15% of the maximum value, the signal to noise at these wavelengths is 0.15 what it is at maximum intensity. The maximum possible signal-to-noise ratio with the detector is 300 at peak intensity. The experiment was always carried out having ensured the maximum count within a reflectance spectrum was about 20% below the saturation level. Thus, the effective maximum signal to noise in a spectrum is approximately 240, and this is reduced to a maximum of approximately 36 at approximately 300 nm and approximately 700 nm. In practice, the spectra showed non-reproducible results in the wavelength range 650–700 nm which showed a tendency to go from flat to being monotonically increasing in this

range as multiple measurements on different spots on the sample were collected. The results in figures 3 and 4 are those that were obtained reproducibly as long as the zero reflectance and 100% diffuse white reflectance calibration was current and the signal level gave an adequate signal to noise. The modest rise in normalized reflectance for the *P. graeffei* silk in the wavelength range 350–300 nm could be due to low signal to noise but it was observed reproducibly. The sharp peak that occurs just below 500 nm, in three of the spectra, is due to drift in the light source intensity which manifests at the wavelength of highest intensity from the light source as an aberrant 100% reflectance normalization. The peak at approximately 530 nm has the same origin as the next highest peak in the spectrum of the light source. These peaks could be removed, or the spectra could be further smoothed, but we present the results as obtained in order to introduce the experimental issues that can arise.

The spectra in figures 3 and 4 are robust and reproducible results obtained by the standard photoreflectance measurement technique applied. If they are interpreted as giving valid information on the reflectance of these four silks, they suggest that three of the silks (*N. plumipes*, *E. transmarina* and *A. keyserlingi*) have reduced reflectance in the UV. This reduction extends into blue for *N. plumipes* and even further for *A. keyserlingi*. *Phonognatha graeffei* has a flat reflectance spectrum, given that it is uncertain that the increase between 350 to 300 nm is real because of a possible signal-to-noise issue. But, it is at this point that the optical form of the sample needs to be brought into the interpretation of the results. As stated in the introduction, there is no physical reason for a dielectric to have wavelength-dependent reflectance variations of the magnitude observed. The light that is collected as the 'reflected' signal has been scattered by the shaped silks, and this includes light that has propagated through the 'largely transparent' silk and ultimately been backscattered to the collection fibre (figure 2*c* and figure 1). Thus, two physical mechanisms contribute to a reduced signal at shorter wavelengths. (i) Scattering—it is known that shorter wavelength light is scattered more than longer wavelengths in Rayleigh scattering within a dielectric, and the form of the silks as a scatterer can lead to a differentiated collection of scattered light at shorter wavelengths compared to longer wavelengths within the numerical aperture of the collection optics (figure 1) [28,29]. (ii) Wavelength-dependent absorption of some of the light by the silk could be occurring. Thus, the reduced signal is considered real but it is attributed to wavelength-dependent absorption and/or wavelength-dependent scattering effects, and not reflection.

The most significant new insight from this study is that the dragline silks of different species of orb weaving spiders have different absorption and scattering properties as determined by this simple, standard measurement technique. In order to refine the measurement, to be able to separate the effect of absorption from the effect of scattering, then measurements employing an integrating sphere [34] or a double integrating sphere [35] should be completed. However, it will be a significant challenge to achieve the necessary sensitivity with samples like the mostly transparent and small-sized spider silks. But the fact that the differences are due to some combination of the cross-sectional shape and the material of the silks is significant.

Comparing the results of this study with those from [19], we note that the shortest wavelength used there was 350 nm compared with 300 nm in this study. With a cut-off wavelength of 350 nm, the normalized photoreflectance for *E. transmarina* is flat at about 1.0 from 650 to 350 nm (figure 3). The sharp reduction in normalized photoreflectance signal for this silk which occurs at 322 nm is a new observation for a spider silk. In [19], the normalized photoreflectance value for *N. clavipes* and *A. argentata* silk started to drop at a longer wavelength and dropped faster than observed for any of the silks in this study. In that study, the normalized reflectance of *M. schreibersi* was reported as having dropped to approximately 0.28 at 390 nm from 1.0 at approximately 560 nm (and longer wavelengths). The normalized reflectance for *N. clavipes* dropped to approximately 0.47 at 350 nm from a value of approximately 0.90 at 500 nm [19]. The analogue *N. plumipes* drops to 0.67 at 350 nm (0.70 at 300 nm) from 1.0 at 500 nm (figure 3). The normalized reflectance for *A. argentata* dropped to approximately 0.40 at 350 nm from a value of approximately 1.0 at 500 nm [19]. The analogue *A. keyserlingi* drops to 0.78 at 350 nm (0.73 at 300 nm) from 1.0 at 500 nm (figure 3). The results for the two *Nephila* species are reasonably close to each other. A 38% drop in normalized reflectance over the 150 nm range from 500 to 350 nm for *N. clavipes* [19] is compared to a 33% drop for *N. plumipes* (figure 3) initiated at longer wavelength. The results for the two *Argiope* species are a 60% relative drop in normalized reflectance for *A. argentata* [19] as compared with a 22% drop for *A. keyserlingi* (figure 3) over the same wavelength range. The photoreflectance signal for *A. keyserlingi* continues to increase at wavelengths longer than 500 nm (figure 3), whereas *A. argentata* had a flat normalized reflectance value beyond this wavelength [19]. The normalized reflectance for *N. clavipes*, *A. argentata* and *M. schreibersi* are all decreasing at a greater rate at shorter wavelengths [19] than observed for any silk in the current study. But, broadly the spectra are similar. It is possible that the larger drops in reflectance

measured in [19] are, at least in part, an artefact of the method used. The signal can drop because shorter wavelengths are being scattered into a larger range of angle, not all of which is collected within the numerical aperture of the collection system.

As absorption by the silk may be playing a role in the measured normalized photoreflectance spectra the amino acid content of the proteins of MA silk may be relevant. Five amino acids—alanine, glycine, glutamine, serine and proline—have been measured to represent approximately 90% of the total amino acids in the MA silks of most spiders [36]. The percentage of these amino acids has been measured for the MA spidroins of the four species in this study [37]. For well-nourished spiders, these amino acids account for approximately 98% (*A. keyserlingi*), approximately 86% (*N. plumipes*), 85% (*E. transmarina*) and 87% (*P. graeffei*) of the total amino acid content [37]. All four have a similar percentage of glycine ranging from (39.2 (0.2))% for *A. keyserlingi* to (43.2 (2.3))% for *P. graeffei*. *Eriophora transmarina* has a marginally lower percentage of alanine ((24.4 (1.0))% compared with up to (33.4 (1.7))% for *N. plumipes*). The MA spidroins of *A. keyserlingi* have significantly more glutamine (11.6 (0.3))% and proline (10.3 (3.0))% than the others. *Eriophora transmarina* and *P. graeffei* have approximately 6.0% and approximately 5.5% of proline, respectively, while *N. plumipes* has only (1.8 (1.5))%. The absorption spectra of the individual amino acids are not well known. Overall, there is no systematic correlation, or set of correlations, between the sequence of the percentages of these five amino acids for the four spider species and the different normalized photoreflectance spectra. Also, it is not possible to draw conclusions with up to 15% of the amino acid content to be attributed to amino acids not specifically isolated and measured to date. It has been established that the percentages of these five amino acids change significantly when the spiders have been deprived of protein in their diet [37]. This may provide the means to make and test a hypothesis linking amino acid content and normalized photoreflectance spectra in future work. But it is also likely that it will be the proteins rather than the constituent amino acids that link to the differentiated photoreflectance spectra. If it is accepted that at least some of the reduced photoreflectance signal as a function of wavelength is associated with absorption then speculation on any relative biological advantage of this can be made. It has been noted that *A. keyserlingi* refreshes its orb web almost daily and UV/light damage of the proteins could be a possible factor in why this is necessary. Low UV absorption of spider silk could be a factor in increasing the useful life of a web.

The silk of *P. graeffei* is neither differentially scattering nor differentially absorbing the light for any subset of the range of wavelength measured. From current knowledge of the shapes of the silks, there is no reason to ascribe this difference as being shape dependent. This means it is a very interesting material to study further to test whether it is a highly UV transparent dielectric. The transmission as a function of wavelength is a key specification of an optical material and the possibility that specific self-assembled proteins might have more favourable UV transmission spectra is intriguing and may prove technically useful in the future following further research. Spider MA spidroin-based silks have been measured to contain proteins with $\beta$ sheet nanocrystals embedded in an amorphous, primarily $\alpha$ helix phase [38,39]. At least some component of the differences in the spectra for the four species measured here is likely to be linked to differences in this nanocomposite nature of the material and the sequences of the proteins.

## 4. Conclusion

A standard experimental technique for measuring the reflectance spectrum of diffuse surfaces, using commercially available equipment, has been applied to specially prepared samples of the dragline silks of four different species of ecribellate orb web spiders. This experiment is relatively similar to a previous experiment [19] that reported the reflectance of silks from a much larger range of species of spiders. By carefully considering the silks as an optical element, it is shown that what is measured is a combination of the relative wavelength dependence of absorption and scattering. Reflectance of the silks is not measured by the method. This changes the significance of the findings and establishes this simple experiment as one that can be used to prospect for spider silks with preferred optical properties such as those with the potential to have low UV absorption. The results for four ecribellate spider species, *P. graeffei*, *E. transmarina*, *N. plumipes* and *A. keyserlingi*, have been reported. From first to fourth, these silks go from not showing any effects of absorption and/or differential scattering down to 300 nm, to showing a greater range of wavelengths over which an increased effect of absorption and differential scattering is observed. Finally, for *A. keyserlingi*, an increasing effect of absorption and differential scattering is observed from 650 to 300 nm. The combined effect of

absorption and differential scattering reduces the signal level by 30–40% at UV wavelengths compared to longer wavelengths, so it is a significant effect in the three silks for which it has been observed. *Phonognatha graeffei* dragline silk emerges as a very interesting spider silk for further investigation with the potential to be demonstrated as a low UV absorbing dielectric micro-fibre.

The results herein give motivation to undertake more detailed comparative study of the silk proteins of the four species studied, particularly with potential relevance to protein optics. With a very large number of spider species producing dragline silk with potentially interesting optical properties, the simple experimental technique used here can be used more widely for prospecting large numbers of different silks to find any which have properties that make them candidates for more detailed study.

Ethics. Not applicable for the organisms used.

Data accessibility. The datasets supporting this article have been uploaded as part of the electronic supplementary material.

Authors' contributions. D.M.K. and S.J.B. conceived the experiment; S.J.B. prepared the spider silk samples; T.E.W. advised on the measurement method; D.J.L. completed the light scattering simulations; D.M.K. carried out the measurements, analysed the results and wrote the first draft of the manuscript. All authors contributed their discipline/topic expertise to the manuscript, critically reviewed the manuscript and gave final approval for publication.

Competing interests. We have no competing interests.

Funding. This research was supported by an Australian Research Council DECRA Fellowship (DE140101281) and a Hermon Slade Foundation grant (no. HSF17/6) to S.J.B.

Acknowledgements. A/Prof. Darrell Kemp of the Department of Biological Sciences, Macquarie University, is thanked for making the photoreflectance equipment available for the experiments.

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
