## [Reviewer comments · Royal Society Open Science]

Review History

RSOS-192174.R0 (Original submission)

Review form: Reviewer 1

Is the manuscript scientifically sound in its present form?

Yes

Are the interpretations and conclusions justified by the results?

Yes

Is the language acceptable?

Yes

Do you have any ethical concerns with this paper?

No

Have you any concerns about statistical analyses in this paper?

No

Recommendation?

Accept as is

Comments to the Author(s)

I have reviewed a prior submission of this manuscript. As stated before, I find a study that uses a standard optical measurement technique to characterize the visibility of spider silk interesting and important, as spider silk plays a major role in spider evolution and diversification, while its visual properties are understudied.

Despite a number of considerations from myself and other reviewers, in my opinion, none of these were of the nature that would undermine the strength of this study. The authors addressed them carefully and I think the manuscript is now in a publishable form.

Review form: Reviewer 2**Is the manuscript scientifically sound in its present form?**

Yes

Are the interpretations and conclusions justified by the results?

Yes

Is the language acceptable?

Yes

Do you have any ethical concerns with this paper?

No

Have you any concerns about statistical analyses in this paper?

Yes

Recommendation?

Accept with minor revision (please list in comments)

Comments to the Author(s)

This version of the manuscript is much improved over the first version. I think that the data and methodology are worth publishing and it seems appropriate for this journal. I have a number of minor comments.

- Some, but not all of the reference to "ancient vs. modern" spiders has been changed. Again, it would be more appropriate to refer to more and less recently derived groups.

- Through most of the manuscript the authors refer to the silk they are studying as "dragline or frame/radial" then later they start referring to this silk as major ampullate or MA silk, then not until page 7 ln 24 is the abbreviation actually defined. Readers familiar with the spider silk literature will know what is going on here, but it would be better to add an explanation in the introduction that explicitly states that spiders use secretions from the major ampullate (MA) glands to produce dragline, frame, and radials.

p3, ln14 "of" should be added between "silks" and "some"

p3, para1 could use some more references

p3, ln 9 ", " should be added between "measurements" and "but"

p3, ln 23 you say "conspecific", but I think you mean "congeneric" unless these are really the same species?

p3, ln 58 "AR" is not defined

p5, ln 7 I don't know what "between" is referring to

Section 3.2 my understanding is that you are using a single strand, from a single spinneret, although as you state, the silk functions as paired strands. Do you have an expectation for how this might affect results? Or an explanation for why you made this choice?

p5, ln 60 is this pattern because it is a big spider and therefore produces big silk?
p6, ln 49-56 could use some references for the optical explanations.

Decision letter (RSOS-192174.R0)

16-Mar-2020

Dear Miss Kane

On behalf of the Editors, I am pleased to inform you that your Manuscript RSOS-192174 entitled "Photoreflectance/Scattering Measurements of Spider Silks Informed by Standard Optics" has been accepted for publication in Royal Society Open Science subject to minor revision in accordance with the referee suggestions. Please find the referees' comments at the end of this email.

The reviewers and handling editors have recommended publication, but also suggest some minor revisions to your manuscript. Therefore, I invite you to respond to the comments and revise your manuscript.

- Ethics statement

- Data accessibility

<http://datadryad.org/submit?journalID=RSOS&manu=RSOS-192174>

- Competing interests

- Authors' contributions

- Acknowledgements

- Funding statement

Because the schedule for publication is very tight, it is a condition of publication that you submit the revised version of your manuscript before 25-Mar-2020. Please note that the revision deadline will expire at 00.00am on this date. If you do not think you will be able to meet this date please let me know immediately.

- 1) A text file of the manuscript (tex, txt, rtf, docx or doc), references, tables (including captions) and figure captions. Do not upload a PDF as your "Main Document";
- 2) A separate electronic file of each figure (EPS or print-quality PDF preferred (either format should be produced directly from original creation package), or original software format);
- 3) Included a 100 word media summary of your paper when requested at submission. Please ensure you have entered correct contact details (email, institution and telephone) in your user account;
- 4) Included the raw data to support the claims made in your paper. You can either include your data as electronic supplementary material or upload to a repository and include the relevant doi within your manuscript. Make sure it is clear in your data accessibility statement how the data can be accessed;

5) All supplementary materials accompanying an accepted article will be treated as in their final form. Note that the Royal Society will neither edit nor typeset supplementary material and it will be hosted as provided. Please ensure that the supplementary material includes the paper details where possible (authors, article title, journal name).

If your manuscript is newly submitted and subsequently accepted for publication, you will be asked to pay the article processing charge, unless you request a waiver and this is approved by Royal Society Publishing. You can find out more about the charges at <https://royalsocietypublishing.org/rsos/charges>. Should you have any queries, please contact openscience@royalsociety.org.

on behalf of Miles Padgett (Subject Editor)
openscience@royalsociety.org

Associate Editor Comments to Author:
Comments to the Author:

Thank you for taking such care in the revision of this transferred manuscript - the reviewers are broadly satisfied that you have taken the requested changes into consideration; however, a number of tweaks remain necessary to get the paper over the line. Please take the same degree of care with these outstanding changes, and we'll look forward to receiving the final version of the paper in the near future.

Reviewer comments to Author:
Reviewer: 1

Comments to the Author(s)

I have reviewed a prior submission of this manuscript. As stated before, I find a study that uses a standard optical measurement technique to characterize the visibility of spider silk interesting and important, as spider silk plays a major role in spider evolution and diversification, while its visual properties are understudied.

Despite a number of considerations from myself and other reviewers, in my opinion, none of these were of the nature that would undermine the strength of this study. The authors addressed them carefully and I think the manuscript is now in a publishable form.

Reviewer: 2

Comments to the Author(s)

This version of the manuscript is much improved over the first version. I think that the data and methodology are worth publishing and it seems appropriate for this journal. I have a number of minor comments.

- Some, but not all of the reference to "ancient vs. modern" spiders has been changed. Again, it would be more appropriate to refer to more and less recently derived groups.

- Through most of the manuscript the authors refer to the silk they are studying as "dragline or frame/radial" then later they start referring to this silk as major ampullate or MA silk, then not until page 7 ln 24 is the abbreviation actually defined. Readers familiar with the spider silk literature will know what is going on here, but it would be better to add an explanation in the introduction that explicitly states that spiders use secretions from the major ampullate (MA) glands to produce dragline, frame, and radials.

p3, ln14 "of" should be added between "silks" and "some"

p3, para1 could use some more references

p3, ln 9 ", " should be added between "measurements" and "but"

p3, ln 23 you say "conspecific", but I think you mean "congeneric" unless these are really the same species?

p3, ln 58 "AR" is not defined

p5, ln 7 I don't know what "between" is referring to

Section 3.2 my understanding is that you are using a single strand, from a single spinneret, although as you state, the silk functions as paired strands. Do you have an expectation for how this might affect results? Or an explanation for why you made this choice?

p5, ln 60 is this pattern because it is a big spider and therefore produces big silk?

p6, ln 49-56 could use some references for the optical explanations.

Author's Response to Decision Letter for (RSOS-192174.R0)

See Appendix A.

Decision letter (RSOS-192174.R1)

30-Mar-2020

Dear Miss Kane,

It is a pleasure to accept your manuscript entitled "Photoreflectance/Scattering Measurements of Spider Silks Informed by Standard Optics" in its current form for publication in Royal Society Open Science. The comments of the reviewer(s) who reviewed your manuscript are included at the foot of this letter.

You can expect to receive a proof of your article in the near future. Please contact the editorial office (openscience_proofs@royalsociety.org) and the production office

(openscience@royalsociety.org) to let us know if you are likely to be away from e-mail contact -- if you are going to be away, please nominate a co-author (if available) to manage the proofing process, and ensure they are copied into your email to the journal.

on behalf of Prof Miles Padgett (Subject Editor)
openscience@royalsociety.org

Appendix A

Anita Kristiansen
Editorial Coordinator

on behalf of Miles Padgett (Subject Editor)
openscience@royalsociety.org Response to Reviewers

25 March 2020

Dear Anita and Prof Padgett,

Thank you for the the notification that, subject to completing minor revisions requested by one of the reviewers, our manuscript will be acceptable for publication in RSOS. Please see a detailed list of Reviewer 2's comment and our response below.

We look forward to the next stages of the publication process.

Yours sincerely,

(Electronic signature)

Deb Kane (on behalf of all authors)

Response to reviewers

Reviewer 1 did not request any revisions.

Reviewer 2

- Some, but not all of the reference to "ancient vs. modern" spiders has been changed. Again, it would be more appropriate to refer to more and less recently derived groups.

There are two uses of "modern" and one use of the term "ancient" in the revised manuscript:

"A prior study of the reflectance of spider silks from 18 species reported results for three species of **modern** orb weaving spiders (*Nephila clavipes*, *Argiope argentata* and *Micrathena Schreibersi*) as having reduced reflectance in the UV range. **(Modern in the context used here means more recently derived).** The reduced UV reflectance was interpreted as an adaptive advantage in making the silks less visible to insects. Herein a standard, experimental technique for measuring the reflectance spectrum of diffuse surfaces, using commercially available equipment, has been applied to samples of the silks of four **modern** species of orb weaving spiders: *Phonognatha graeffei*; *Eriophora transmarina*; *Nephila plumipes*; and *Argiope keyserlingi*."

“Nonetheless if silk visibility by day is the selective driving silk reflectance, we might expect *E. tramsmarina* silk to be slightly more reflective in the UV than that of *A. keyserlingi*. *Phonognatha graefei*, or leaf curling spider, is reported as a nocturnally active web building spider of the family Araneidae (but formerly classed with the more ancient family Tetragnethidae).”

In these 3 instances the word is chosen to link with published work being referred to and/or to reflect common usage description that will be clear to most readers. On balance we think the word choice, as it currently is, will be more readily understood by more readers. We nevertheless added that “modern in the context used here means more recently derived” to show that we use it in the same evolutionary context that the reviewer has interpreted it.

- Through most of the manuscript the authors refer to the silk they are studying as "dragline or frame/radial" then later they start referring to this silk as major ampullate or MA silk, then not until page 7 In 24 is the abbreviation actually defined. Readers familiar with the spider silk literature will know what is going on here, but it would be better to add an explanation in the introduction that explicitly states that spiders use secretions from the major ampullate (MA) glands to produce dragline, frame, and radials.

The first use of the term major ampullate silks was on page 2 of the submitted manuscript. In response to the reviewers comment we have changed page 2 line 7 to include major ampullate and introduced (MA) in page 2 paragraph 4 line 1.

The species *Argiope keyserlingi*, and *Plebs eburnus* have been studied and shown to have high optical quality dragline/radial (i.e. major ampullate) silks [11-17].

“In this study we measured the photoreflectance/scattering of major ampullate (MA) silks from four species of spider:”

p3, In14 "of" should be added between "silks" and "some"

Corrected “The study examined the silks of some derived ecribellate species”

p3, para1 could use some more references

Without some more specific suggestion of what referencing is missing from this paragraph we are unsure of what the reviewer thinks is missing. This paragraph deals with the actual previous measurements of the optical properties of silks that are relevant to this paper. There is a literature on measuring birefringence of spider silks, which we have mentioned and have included references to, but that was done for completeness. That research is not directly relevant here. We don't think we have missed any references that should be included.

p3, In 9 ", " should be added between "measurements" and "but"

Corrected

p3, In 23 you say "conspecific", but I think you mean "congeneric" unless these are really the same species?

The term we should have used is 'congener'. We have thus changed the text in question to:

"*N. plumipes* is a congener of *N. clavipes* and *A. keyserlingi* is a congener of *A. argentata*"

p3, In 58 "AR" is not defined

Corrected "Such a layer could act as an antireflection (AR) coating with a centre wavelength determined by the thickness of the layer. The centre wavelength could be in the UV. The aqueous layer on a capture silk, could potentially form an AR coating, for example."

p5, In 7 I don't know what "between" is referring to

The paragraph has been changed to make the meaning clear.

"We collected six females (identified visually) of each of the four species of spider (*N. plumipes*, *E. transmarina*, *A. keyserlingi*, and *P. graeffei*, totalling 24 spiders) from locations in suburban Sydney, and taken to the laboratory at the University of New South Wales in eastern Sydney for silk collection. The six were a combination of non-gravid adult and sub-adult spiders.

Section 3.2 my understanding is that you are using a single strand, from a single spinneret, although as you state, the silk functions as paired strands. Do you have an expectation for how this might affect results? Or an explanation for why you made this choice?

The experiment uses a single thread from the spinnerets so the silk sample used is a double cylinder thread. We have in separate studies undertaken theoretical modelling of scattering from a double cylinder and compared it to a single cylinder. Models of scattering from a double cylinder (both ours and the one referenced [18]) involve some approximation and are not exact. The scattering from a double cylinder changes the detail of the maxima and minima in the scattering pattern with angle but it does not make a significant difference to the angular spread of the main lobe of the scattering pattern (both forward and back scattered). Hence, we have not obfuscated the main narrative here with, what in the context of the paper, is unnecessary detail, by discussing scattering from a double cylinder in any more detail.

p5, In 60 is this pattern because it is a big spider and therefore produces big silk?

Yes, it is primarily because the silks have the largest diameter. We have added a sentence to make this explicit.

"It is noted that it is the silks of this nocturnal species that has the largest photoreflectance signal per silk in this study. This is primarily because of their larger diameter."

p6, In 49-56 could use some references for the optical explanations.

The eight lines of text associated with this comment are:

“The light that is collected as the “reflected” signal has been scattered by the shaped silks, and this includes light that has propagated through the “largely transparent” silk and ultimately been back scattered to the collection fibre (fig. 2(c) and fig. 1). Thus, two physical mechanisms contribute to a reduced signal at shorter wavelengths. (i) Scattering – it is known that shorter wavelength light is scattered more than longer wavelengths in Rayleigh scattering within a dielectric, and, the form of the silks as a scatterer can lead to a differentiated collection of scattered light at shorter wavelengths compared to longer wavelengths within the numerical aperture of the collection optics (Fig. 1) [28, 29]. (ii) Wavelength dependent absorption of some of the light by the silk could be occurring. Thus, the reduced signal is considered real but it is attributed to wavelength dependent absorption and/or wavelength dependent scattering effects, and not reflection.”

We have added references to this section as requested.